# The Effect of CFTR Modulators on Airway Infection in Cystic Fibrosis

**DOI:** 10.3390/ijms23073513

**Published:** 2022-03-23

**Authors:** Caitlyn Harvey, Sinead Weldon, Stuart Elborn, Damian G. Downey, Clifford Taggart

**Affiliations:** 1Airway Innate Immunity Research (AiiR) Group, Wellcome-Wolfson Institute for Experimental Medicine, Queen’s University Belfast, Belfast BT9 7BL, UK; charvey10@qub.ac.uk (C.H.); s.weldon@qub.ac.uk (S.W.); 2School of Medicine, Dentistry and Biomedical Sciences, Queen’s University Belfast, Belfast BT9 7BL, UK; s.elborn@qub.ac.uk; 3Wellcome-Wolfson Institute for Experimental Medicine, Queen’s University Belfast, Belfast BT9 7BL, UK; d.downey@qub.ac.uk; 4Belfast Health and Social Care Trust, Belfast BT13 1FD, UK

**Keywords:** cystic fibrosis, CFTR modulator, airway infection

## Abstract

The advent of Cystic fibrosis transmembrane receptor (CFTR) modulators in 2012 was a critical event in the history of cystic fibrosis (CF) treatment. Unlike traditional therapies that target downstream effects of CFTR dysfunction, CFTR modulators aim to correct the underlying defect at the protein level. These genotype-specific therapies are now available for an increasing number of CF patients, transforming the way we view the condition from a life-limiting disease to one that can be effectively managed. Several studies have demonstrated the vast improvement CFTR modulators have on normalization of sweat chloride, CFTR function, clinical endpoints, and frequency of pulmonary exacerbation. However, their impact on other aspects of the disease, such as pathogenic burden and airway infection, remain under explored. Frequent airway infections as a result of increased susceptibility and impaired innate immune response are a serious problem within CF, often leading to accelerated decline in lung function and disease progression. Current evidence suggests that CFTR modulators are unable to eradicate pathogenic organisms in those with already established lung disease. However, this may not be the case for those with relatively low levels of disease progression and conserved microbial diversity, such as young patients. Furthermore, it remains unknown whether the restorative effects exerted by CFTR modulators extend to immune cells, such as phagocytes, which have the potential to modulate the response of people with CF (pwCF) to infection. Throughout this review, we look at the potential impact of CFTR modulators on airway infection in CF and their ability to shape impaired pulmonary defences to pathogens.

## 1. Introduction

Airway infections involving the lower respiratory tract are estimated to claim the lives of around 4 million individuals worldwide each year [1]. People with cystic fibrosis (pwCF) have a particular heightened susceptibility to such airway infections, which are known to be a major contributor to their morbidity and mortality [2]. Cystic fibrosis (CF) is a life-limiting, autosomal recessive disorder resulting from mutations in the CF conductance regulator gene located on the long arm of chromosome 7 at position 7q31 [3,4]. The gene codes for the cystic fibrosis transmembrane regulator protein (CFTR) [4,5], a 1480 amino acid cyclic-AMP (c-AMP) activated transmembrane protein that controls the secretion of chloride (Cl^−^) and bicarbonate (HCO_3_^−^) ions across epithelial barriers, thereby regulating the composition of mucosal fluids [6,7]. In addition, CFTR modulates the activity of the sodium (Na^+^) epithelial channel (ENaC), dysfunction of which results in impaired innate immune responses [8]. To date, over 2100 CFTR mutations have been described, of which around 380 are known to cause disease [9]. These mutations are of varying severities, ranging from complete absence of protein to partial dysfunction [9,10,11]. The CFTR is located on epithelial cells of all mucosal surfaces including the gastrointestinal (GI) tract, kidneys, liver, reproductive tract (males), and the lungs, resulting in multiple organ dysfunction in pwCF [5,12,13,14]. However, the main cause of morbidity and mortality in CF occurs within the respiratory system. Aberrant ion transport due to dysregulated CFTR function impairs mucociliary clearance (MCC) and, subsequently, pwCF experience recurrent and severe airway infections contributing to the development of a hyperinflammatory state [14,15,16]. Chronic inflammation within the lung leads to decline in lung function, which, over time, ultimately results in respiratory failure and premature death. Traditional therapies target the downstream effects of CFTR dysfunction and include symptom management through airway clearance and, the prevention and treatment of infection with antimicrobials. However, due to the pivotal role of inflammation in lung damage, development of anti-inflammatories has been an area of intensive research [17]. Despite their pre-clinical efficacy in dampening inflammation [17,18], these treatments are non-corrective and, other than ibuprofen [19,20,21], have not migrated into the clinical space due to the risk of reducing an already impaired host response in chronically ill patients, potentially increasing exacerbations [22,23,24]. Alternative, more efficient lines of treatment that target the basic defect of CFTR dysfunction are, therefore, essential for the effective management of the condition.

The development of CFTR modulators within the last decade has transformed the treatment of CF [25,26]. Rather than targeting the downstream consequences of CFTR dysfunction, they aim to partially correct the actual CFTR defect at the protein level. High throughput screening analysis of gating mutations identified the G551D mutation as the most common; hence, it was the first of the six classes of mutation to be targeted. The small molecule drug (Ivacaftor [IVA]) can be used to potentiate the normal function of the protein [26,27,28,29]. However, with only 5–8% of the CF population possessing such a mutation, modulators capable of treating a wider cohort of patients were required [28]. The most common mutation is the F508del mutation with around 90% of pwCF carrying at least one copy [30]. It results in misfolding of the CFTR protein, defective trafficking, and, subsequently, a reduced apical expression of functional CFTR. The CFTR modulator Lumacaftor (LUM) was developed as a corrector to improve protein folding [31,32]. Dual combination therapy involving coupling a potentiator and a corrector has been shown to produce modest improvements in lung function. However, this effect was limited to F508del homozygotes [29]. LUM/IVA combinations were found to be effective in partially restoring gating mutations; however, it was reported to cause respiratory side effects [16]. As a result, the corrector tezacaftor (TEZ) was developed as an alternative. The efficacy of TEZ was reported to be relatively similar to LUM; however, it produced a more favourable side-effect profile. In 2019, triple therapy, a combination of two correctors (elexacaftor and tezacaftor) and one potentiator ivacaftor, commercially known as Kaftrio (ETI), was approved in the US for the effective treatment of CF, resulting in a 63% decrease in pulmonary exacerbations (PEx) and significant improvements in lung function for both homozygotes and heterozygotes with the F508del mutation [29,33,34]. The definitive mechanism of action of CFTR modulator potentiators (IVA) is unknown; however, it is thought to restore the normal movement of Cl^-^ through the epithelial cell membrane [35]. Correctors such as LUM and TEZ mimic the action of protein folding chaperone proteins, preventing misfolding and destruction at the endoplasmic reticulum, resulting in increased CFTR apical expression [36]. Whilst CFTR modulators have revolutionised the treatment of CF by improving CFTR expression, function, and quality of life, the impact on other aspects of the condition such as disease progression, pathogenic burden, and airway infection is conflicting [10,37]. Throughout this review we analyse the microbiome and origins of airway infection within the CF lung and explore the potential impact of CFTR modulators to regulate the microbial landscape and reduce the susceptibility to airway infection.

## 2. Infection in CF

Repeated vicious cycles of infection and inflammation lead to progressive tissue damage in CF. A culmination of multiple factors leads to an increased susceptibility to infection. The airway surface liquid (ASL) forms part of the innate immune system’s ‘first-line’ of defence, aiding the clearance of microorganisms through maintaining MCC [7,38]. The viscosity and composition of the ASL is controlled through ion transport via the CFTR. Dysfunctional CFTR impairs MCC through alteration of the ASL. In CF, the ASL becomes dehydrated, acidified, and viscous, resulting in a dense, sticky mucus [39,40,41,42]. Consequently, mucus accumulates, forming plugs in the small airways and the low pH impairs ciliary beating and phagocyte function [39,43]. The combination of increased viscosity and impaired MCC provides the perfect opportunity for the colonisation of microorganisms within the airways resulting in persistent and recurring airway infection (Figure 1).

## 3. Microbiome in CF

Culture independent methods, such as 16S ribosomal RNA (rRNA), have revealed a diverse, niche-specific set of microorganisms harboured within the lung [2]. The composition and relative abundance of microorganisms within the CF lung changes in an age dependent manner [44]. CF lungs tend to be dominated by classic opportunistic pathogens such as *P. aeruginosa*, *S. aureus*, *B. epacian* non-typeable *Haemophilus influenzae*, *Stenotrophomonas maltophilia*, and *Achromobacter xylosoxidans* [2,45,46,47,48]. Although different to their non-CF counterparts, the microbial diversity is somewhat preserved in the early CF lung which is colonised with around 50% ‘non-classical’ CF pathogens such as *Prevotella*, *Streptococcus*, *Veillonella,* *Haemophilus*, and Fusobacterium [10]. Of the classical CF pathogens, within the early CF lung, *S. aureus* tends to dominate. As individuals age, the intrinsic predisposition of the CF lung to infection makes long term nebulised antibiotic treatment inevitable. Although effective in alleviating the severity of infection, their long-term use coupled with progressive tissue damage and disease severity results in the selection and accumulation of classic CF pathogens reducing microbial diversity [2,49,50,51]. Diminished microbial diversity is associated with a decline in lung function, more frequent Pex, and poor clinical outcomes [52]. In addition, upon entry and colonisation in the airway, bacteria have the ability to acquire phenotypic changes displaying a shift from a free-living planktonic state to complex sessile community of microorganisms encapsulated in an extracellular matrix (biofilm) (Figure 1) [53]. This occurs through the change in expression of virulence factors, such as type 3 secretion systems [54,55] and flagellum [56,57], both of which are downregulated, along with the secretion of exoproducts to reduce host bacterial killing [58] overall, facilitating and promoting the establishment of chronic pulmonary infections [16,59,60]. Chronic airway infections are a major problem faced by pwCF and are difficult to eradicate, often requiring long-term suppression with inhaled antibiotics and, consequently, increasing the risk of developing further antimicrobial resistance, trapping these patients in a vicious cycle of infection and inflammation.

### Dysregulated Anti-Microbial Responses in CF

Coupled with an increase in susceptibility to infection, pwCF have dysregulated immune and antimicrobial responses to infection. Acquisition of organisms leading to infection triggers a hyperinflammatory state in CF, which is dominated by neutrophils [14,61,62]. Despite the abundance of neutrophils, pwCF fail to successfully clear infection, demonstrating that mutations in the CFTR have direct consequences on host innate immune responses [63,64]. Non/dysfunctional CFTR compromises several aspects of neutrophil function in response to infection. Impaired chlorination results in reduced killing of phagocytosed bacteria, providing opportunity for bacterial colonisation, persistence, and infection [65,66,67]. Absence of CFTR leads to an ion imbalance within the neutrophil (increasing cytosolic chlorine and sodium and reducing magnesium), which has been associated with changes in neutrophil degranulation patterns [68,69,70]. It has been reported that CF neutrophils release more primary granules (such as neutrophil elastase (NE), levels of which negatively correlate with lung function) than secondary and tertiary granules [68,71,72]. In addition, studies have observed an increased abundance of neutrophils possessing a pro-survival phenotype within the CF lung [73]. These neutrophils exhibit reduced apoptosis contributing to the self-perpetuating cycle of inflammation and infection [74,75,76].

Alveolar macrophages (AM) are the primary phagocytic cells of the innate immune system, enabling clearance of debris and infectious pathogens. Previous studies have demonstrated higher baseline numbers of AM in non-infected CF children compared to non-CF children providing evidence of a pre-existing pathogen independent mononuclear inflammation in CF [77,78,79]. Lysosomes from macrophages isolated from CFTR^-/-^ deficient mice fail to reach a pH lower than 5, which impairs the bactericidal activity of the macrophage resulting in an overall defective phagocytic pathway in CF [80]. Macrophages tend to be polarised into two subsets: 1, M1 pro-inflammatory; and 2, M2 anti-inflammatory [81]. There is an increased abundance of M1 macrophages isolated from CF polyps, suggesting a pro/anti-inflammatory imbalance within CF; however, data surrounding AM polarisation in CF are conflicting [82,83,84].

Anti-microbial peptides (AMPs) are ubiquitous among epithelial barriers of mucosal surfaces and form the ‘natural antibiotics’ of the immune system, exhibiting both direct anti-microbial and immunomodulatory activity [85,86]. Several studies using knockout mice have corroborated hypotheses that AMPs play a pivotal role in inhibiting microbial proliferation [87]. They are primarily found in neutrophilic granules and secretions from epithelial surfaces [88]. The presence of an abundant source of AMPs in the CF lung has been reported [89]. However, despite their abundance, pwCF fail to mount an effective anti-microbial response with evidence of impaired bacterial killing [90]. AMPs, such as cathelicidin (LL-37) and secretory leukocyte protease inhibitor (SLPI), are inhibited in CF by the overexuberant production of NE produced as a result of infection in the lung [91,92,93]. In addition, proteinases produced from classic CF pathogens in an attempt to evade host defences have been shown to degrade AMPs, such as LL-37, contributing to the cycle of infection and inflammation [94,95,96]. Thickened airway mucus in CF contains high concentrations of negatively charged glycosaminoglycans and extracellular DNA which have the potential to bind AMPs through anionic/cationic attractions, rendering the AMP non-functional [96,97], and further increasing the vulnerability of the CF lung to infection. The ability of bacteria to form highly hydrophobic negatively charged biofilms within the CF lung [98] provides microorganism protection from AMPs and some antibiotics due to the repellent charge they possess, helping to facilitate the establishment of chronic infections in CF. Overall, acute, and chronic airway infection is a major contributor to morbidity, mortality, and low quality of life of individuals with CF; therefore, exploration into novel eradication measures is required.

## 4. Impact of CFTR Modulators on Airway Infection

Clinical trials focus on the effect of CFTR modulators on pulmonary function (absolute change from baseline in percentage predicted FEV_1_), rate of pulmonary exacerbation, change in sweat chloride, and the change in quality of life [25,28,32,33,34,99]. However, there are knowledge gaps in how CFTR modulators impact airway physiology and infection and their ability to modulate the susceptibility to infection. Much of the published data is limited to the impact of IVA with more limited data on dual or triple CFTR modulators. Some evidence suggests that CFTR modulators have direct anti-microbial activity against the bacterial pathogens *S. aureus* and *Streptococcus pneumonia* [100,101], which may reduce bacterial load and pathogenic burden. Combined modulator treatment (TEZ/IVA) of children with CF has been shown to restore bacterial diversity to a similar composition as their non-CF age matched controls [23]. Similar responses have been observed in adult cohorts with a GD155 mutation [102,103]. Twenty nine percent of the participants who enrolled within the GOAL study, who were initially culture positive for *P. aeruginosa*, became culture negative following a year of IVA treatment [47]. Such positive findings suggest that CFTR modulation induces a shift in host-pathogen interactions that may be beneficial for the composition of the airway microbiome reducing the incidence of acute airway infection and slowing the rate of lung decline. Mechanisms underpinning this phenomenon are yet to be elucidated. However, literature is conflicting with contrasting studies showing that despite diminishing *P. aeruginosa* culture positivity, treatment with potentiator IVA does not reduce the odds of culture positivity with *S. aureus* or other common CF pathogens, such as *H. influenza*, *S. maltophilia* or *aspergillus spp*. [47]. In addition, it has been reported that the presence of bacterial pathogens such as *P. aeruginosa* limit the efficiency of CFTR modulators through the production of exoproducts that have the ability to diminish the rescue of CFTR by correctors VRT-325 and LUM [104,105], thus, contributing to lung function decline through exacerbating disease pathology.

Structurally, IVA resembles quinolone antibiotics. A quinolone ring is central to the molecule and so it has been suggested that it may have direct anti-bacterial properties towards certain pathogens through disruption of bacterial DNA replication [106]. IVA has been demonstrated to exert bactericidal activity towards CF clinical isolates of *Streptococcus* spp. and bacteriostatic activity against *S. aureus* [107]. Reznikov et al. also observed a dose dependent reduction in both *S. aureus* and *P. aeruginosa* bioluminescence following treatment with IVA, indicating a decrease in the abundance of bacteria [100]. This reduction was comparative to the reduction in bacterial growth observed when treated with vancomycin suggesting that treatment of CF patients with IVA may mitigate the susceptibility to airway infection through direct killing of classic CF pathogens in the lung. Contrasting evidence show the inhibition of DNA gyrase by IVA is weak and limited [108]; however, this is likely due to the inability of IVA to cross the highly hydrophobic gram-negative bacterial membrane. Interestingly, antibiotic/CFTR modulator combinations display a synergistic killing effect against *P. aeruginosa* isolates reducing bacterial counts (CFU/mL) by 100-fold [108]. Positive interactions have also been observed with ceftriaxone, lineoid, ciprofloxacin, and vancomycin, which have demonstrated efficacy in reducing growth of S*. aureus, S. pneumoniae*, and *P. aeruginosa* [100,101,107,109]. In addition, LUM has been shown to promote the production of reactive oxygen species [108], which can directly kill bacteria through inducing oxidative stress [110]. It is plausible to suggest that antibiotic treatment permeabilises the bacterial membrane, enabling the entry of CFTR modulators, where they can exert a direct antimicrobial effect potentially reducing pathogenic burden within CF airways and reducing infection. A key virulence factor employed by many bacteria is the presence of efflux pumps, which actively remove antibiotics preventing bacterial killing. CFTR modulators function by improving the activity of ion channels, therefore, it is feared that they may also improve function of bacterial ion channels, increasing the removal of antimicrobials contributing to infection [111]. However, to date, evidence suggests that there is no negative relationship between CFTR modulators and antibiotics with in vivo studies demonstrating no change in antibiotic susceptibility of *P. aeruginosa* when treated in combination with IVA [112,113], indicating it has no damaging effects on microbiological parameters and thus airway infection.

However, a major bottleneck to the current evidence is that most studies have been carried out in vitro. Evidence of the presence of IVA in the sputa of pwCF treated with IVA is limited. Ex vivo analysis of overnight IVA concentrations within CF sputum have demonstrated small concentrations of IVA (0.10 ± 0.03 µg/mL) are present [114]. According to in vitro studies an optimum concentration of greater than 1µg/mL is required for antimicrobial activity of IVA [109]. It is, therefore, possible that the concentrations CFTR modulators in CF sputum are not sufficient to exert direct antimicrobial activity on luminal bacterial pathogens.

Retrospective observational studies have shown IVA reduces total bacterial concentration in sputum. Most notably, there was a marked reduction in the relative abundance of *P. aeruginosa* within chronically infected individuals after 48hrs of treatment [115]. However, no chronically infected *P. aeruginosa* patients were consistently culture negative and bacterial density gradually began to increase following long-term treatment (1 year) of IVA. Similar results were observed by Durfey et al. who demonstrated a 10-fold decrease in bacterial density of *P. aeruginosa* and *S. aureus*; however, only a small number of patients were consistently culture negative [116]. The data presented in both studies suggest that CFTR modulation may have a direct impact on bacterial killing, leading to a short-term reduction in airway infection. However, the subsequent bacterial increase following long term treatment with IVA indicates a negative impact of CFTR modulators on the lung microbiome; increasing susceptibility to infection and posing the question, can the bacteria deploy similar strategies to antimicrobial resistance to protect themselves from killing by CFTR modulators?

The effect of CFTR modulators is not limited to bacteria. The CF lung is home to a plethora of microbes including fungi and viruses. *Aspergillus fumigatus* is a common fungal pathogen isolated from CF airways [2]. Its presence can lead to the development of allergic bronchopulmonary aspergillosis (ABPA), which has been reported to have a negative impact on lung function in CF [117,118]. Studies utilising registry databases indicate a beneficial role of IVA on the colonisation of *Aspergillus* in the CF lung [119,120], reducing the opportunity for the establishment of airway infection (Table 1).

In addition, evidence obtained from both clinical and in vitro studies suggests antiviral responses in pwCF are impaired [121,122,123,124]. Respiratory syncytial virus (RSV) and Rhinovirus (RV) are two of the most prevalent viral pathogens infecting children [125,126]. However, infection with either of these two pathogens in children with CF is associated with higher viral loads, increased hospitalisations, and a significant decline in lung function [123,127,128]. It is unknown whether treatment with CFTR modulators have the ability to restore dysfunctional anti-viral responses in pwCF reducing the prevalence of airway infection. De Jong et al. found that treatment with CFTR modulators (IVA, IVA/LUM) had no direct impact on the anti-viral response to RV in CF airway epithelial cells at the gene level [129]. The data presented indicate that CFTR modulation has no impact on the control of viral infection within CF airway epithelial cells and thus is unable to reduce susceptibility to airway infection. However, studies looking at the impact of CFTR modulators on the antiviral response in CF are sparse and remain an area of investigation.

Overall, CFTR modulator therapy has a positive effect on the pathophysiology of CF. Correction of CFTR expression and function restores the ion balance, normalising the ASL barrier leading to improved MCC [130]. Improved viscosity, ciliary beating, and mucus clearance will reduce the opportunity for pathogens to colonise the lung, having a potential positive knock-on effect on the acquisition of new organisms and the incidence of acute airway infection. The ASL contains a plethora of AMPs which are rendered ineffective in CF, increasing susceptibility to infection [38,42,91,107]. Normalisation of the pH will produce an optimum environment for the effective functioning of proteases, increasing bacterial and viral killing, and preventing airway infection and the establishment of chronic infections. 

Innate immune cell responses have been reported to be somewhat restored following CFTR modulation. Improving CFTR protein function restores ion imbalance within neutrophils normalising degranulation and consequently improving microbial killing reducing susceptibility to airway infection [67]. In addition, CF neutrophils treated with IVA demonstrated increased apoptosis, potentially aiding the killing of phagocytosed bacteria, and minimising opportunity for bacterial colonisation and infection [131]. However, the mechanisms behind the direct impact of IVA on apoptosis have not been elucidated. There is evidence that AM from pwCF treated with IVA have improved phagocytosis and bacterial killing [132,133,134,135]. Interestingly, this effect was lost when monocyte derived macrophages were treated with combinations CFTR modulator therapy [132]. Therefore, despite the lack of evidence of a direct antimicrobial effect of CFTR modulators, the overall prevalence of classic CF pathogens and incidence of airway infection is likely to decrease due to improvements in innate immune function and the cellular microenvironment. CFTR dysfunction is linked to an impaired resolution of inflammation [136], therefore, such changes may facilitate the cessation of the vicious cycle of inflammation and improve microbial killing, subsequently, reducing the frequency of airway infection in pwCF.

## 5. Conclusions and Future Work 

The limited evidence of a direct impact of CFTR modulation therapy on microbial killing in CF does not necessarily indicate that they are ineffective in preventing airway infection. Many of the studies conducted use small cohorts and, as shifts in established microbial communities often take some time, global longitudinal studies would more accurately represent the impact of CFTR modulators on the airway microbiome and consequently infection. The exact cause of inconsistent responses in regard to microbial clearance and reduced airway infection following CFTR modulation is unknown; however, it is likely caused by heterogeneity in the improvement of MCC between individuals rather than a direct antimicrobial effect. Differential responses confirm that despite a ‘one size fits all’ approach taken when treating with CFTR modulators, it is difficult to predict how these patients will respond, and a personalised approach to each individual may be more effective. Typically, CF is a progressive disease and so older patients often present with more severe structural airway damage and advanced bronchiectasis [15], which, alone, is a risk factor for the establishment of chronic infections in CF [137]. It is unlikely that CFTR modulators are able to reverse such advanced tissue damage. Therefore, it would be rare for the microbial composition of the CF lung to change following treatment with CFTR modulators in these individuals. For this reason, it is essential that CFTR modulator treatment is introduced from an early age before irreversible lung disease is established. With recent approval for the use of single CFTR modulators (IVA) in children aged 2 years, it may be some time before the true potential impact of CFTR modulators on modulating airway infection can be accurately assessed.

Determining the ability of CFTR modulators to control airway infection is an important and emerging area of research. If effective, early intervention of CFTR modulators from infancy may aid the conservation of microbial diversity providing less opportunity for the development of chronic airway infections [138], reducing treatment burden, which could potentially significantly improve the quality of life of pwCF.. To date, data suggest that CFTR modulators have a potential role in the prevention and treatment of airway infections through both direct and indirect mechanisms. However, whether this is generalised or species specific is unknown and inconsistencies in the data warrant further investigation as to whether the effect offers short term infection prevention or sustained. The majority of current studies focus on the investigation of adults with more established lung disease rather than children with little lung damage. However, the recent approval for the use of triple therapy (ETI) in children ages 6–11 will provide a patient population to demonstrate the potential impact of CFTR modulators on airway infection longitudinally.

## Figures and Tables

**Figure 1 ijms-23-03513-f001:**
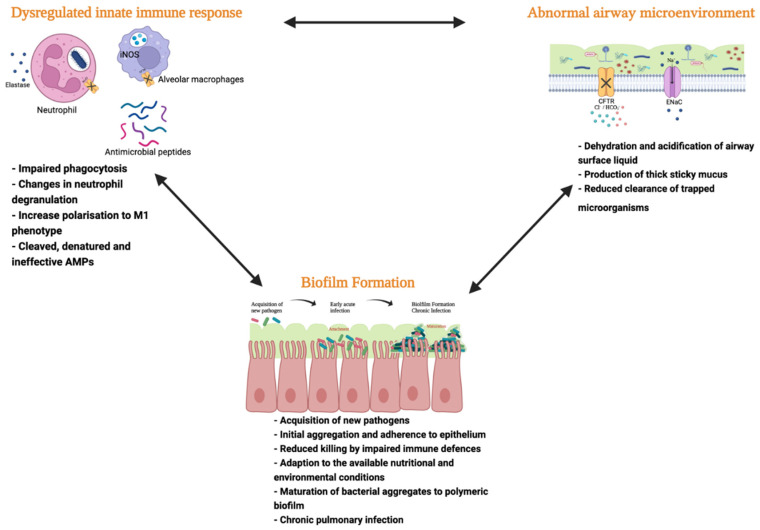
Factors contributing to an increased risk of airway infection in Cystic fibrosis (CF). Aberrant airway microenvironment driven by defective cystic fibrosis transmembrane conductance regulator (CFTR) function results in dehydration and acidification of the airway surface liquid (ASL), impairing mucociliary clearance (MCC), and renders host innate defences ineffective. Acquisition of new microorganisms within the CF lung leads to the development of acute bacterial infections. Defective CF host immune responses result in reduced killing of bacteria, providing an opportunity for bacterial adaption to nutritional and environmental conditions within the airways. Such adaptations may include the downregulation of virulence factors such as flagellin and type 3 secretion systems, enabling a shift from planktonic bacterial infection to biofilm formation. This is particularly common in *P. aeruginosa* infections. Formation of the complex polymeric biofilm matrix makes it very difficult to eradicate these organisms, leading to the development of chronic pulmonary infections. The cumulative impact of the intrinsic defective CFTR function on the airway microenvironment, host innate defences, and the ability of bacterial pathogens to adapt to their environment and grow in a multicellular and sessile form (biofilms) leaves the CF lung highly permissive to airway infection.

**Table 1 ijms-23-03513-t001:** Overview of clinical studies demonstrating an anti-microbial effect of CFTR modulators.

Year	First Author	Title	Effect
2014	Heltshe	*Pseudomonas aeruginosa* in Cystic Fibrosis Patients with G551D-CFTR Treated with Ivacaftor	Reduction of *P. aeruginosa* Culture positivity
2017	Hisert	Restoring Cystic Fibrosis Transmembrane Conductance Regulator Function Reduces Airway Bacteria and Inflammation in People with Cystic Fibrosis and Chronic Lung Infections	Reductions in sputum *P. aeruginosa* density
2017	Chmiel	A double-blind, placebo-controlled phase 2 study in adults with cystic fibrosis of anabasum, a selective cannabinoid receptor type 2 agonist	Anti-microbial
2018	Millar	*Pseudomonas aeruginosa* in cystic fibrosis patients with c.1652G›A (G551D)-CFTR treated with ivacaftor-Changes in microbiological parameters	Reduction of *P. aeruginosa* rate and density
2019	Zhang	Influence of CFTR Modulators on Immune Responses in Cystic Fibrosis	Improved bacterial diversity
2020	Favia	Treatment of Cystic Fibrosis Patients Homozygous for F508del with Lumacaftor-Ivacaftor (Orkambi^®^) Restores Defective CFTR Channel Function in Circulating Mononuclear Cells	Improved bacterial diversity
2021	Durfey	Combining Ivacaftor and Intensive Antibiotics Achieves Limited Clearance of Cystic Fibrosis Infections	Reduction in *S. aureus* positivity

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
