# Peer review of "The Effect of CFTR Modulators on Airway Infection in Cystic Fibrosis"

_ijms, 2022, doi:10.3390/ijms23073513_

Round 1

Reviewer 1 Report

Cystic fibrosis (CF) impairs the synthesis of ion-regulating channel cystic fibrosis transmembrane conductance regulator (CFTR) further contributing to the formation of a bacterial-favored lung microenvironment that increases susceptibility of patients to lung infections. In this review, the authors discuss the microbiome aspect of CF in respect to CFTR modulators. The review starts by introducing the complex of microbiome and immune response in the CF lungs, then the role/impact of CFTR modulators in the network. The discussion point is interesting and overall writing is with unity and coherence.

  1. Authors may elaborate on the modulators’ mechanism of action.
  2. It would give a clearer view if the cited clinical trials were tabulated.
  3. Author may discuss more on the practical anti-bacterial routine in CF treatment.
  4. Minor typing errors at line 66 (management), line 33 (selection), line 247 (evidence), and line 261 (patients).
  5. Punctuation error at line 80, please correct “function,.” to “function.”.
  6. Punctuation error at line 109, please synchronize format as the text description in figure 1 (with or without the space after hyphen).

Reviewer 2 Report

Harvey et.al. provide a review on the effect of CFTR modulator therapy on CF airway infection. The study is generally well developed with only minor weaknesses.

Fig. 1: the bottom panel of Fig. 1 should be revised. Currently, it seems to imply that antibiotic treatment is the cause of (or triggers) biofilm formation and development of chronic infections. Neither is true. Pathogens such as P.aeruginosa form biofilms also in the absence of antibiotics, and chronic airway infections were a hallmark of CF prior to use of antibiotics.

Indeed, in some instances the phrasing of the manuscript (in particular wording related to Fig. 1) leaves the impression as if use of anti-biotics in CF is a “bad” thing and should be discouraged in patients to prevent development of “anti-biotic resistance”, “biofilm formation”, and “chronic infections”. This is not the case as starting a rigorous anti-biotic treatment early in childhood (and perhaps before the first infections occur) was shown to be beneficial to maintaining lung function in the long run. Leaving the impression that anti-biotic use should be discouraged would be unhelpful for the CF community.

The legend to Fig. 1 contains a statement suggesting that it is prolonged antibiotic treatment that promotes virulence factors and leads to biofilm formation. This statement is too broad and should be revised (e.g. consider listing specific virulence factors). There are numerous virulence factors and some of which are downregulated (not upregulated) during adaptation to biofilms (e.g. T3SS effectors or flagellum) and changes in virulence factor expression during adaptation to biofilms also occur in the absence of antibiotics.

Ln118: The statement that culture-independent methods have eradicated the idea of a sterile lung seems overly strong and should perhaps be toned down. In a normal/non-CF subject, the airways do not host a microbiome comparable in sheer numbers to that of the GI tract or other niches (e.g. oral, vaginal, skin). Hence, in comparison, one may want to consider the normal airway relatively sterile. While culture-independent methods do allow detection of pathogens that are difficult to culture/grow in the lab, the flip-side is that culture-dependent methods have the advantage to confirm that the pathogen was “alive” at time of detection.

There is a surprisingly large number of spelling errors and missing commas. Some proofreading (or at least use of the spell check feature of a WORD processing software) is advised.
